# The needs and opportunities of older laypeople to acquire first aid skills

Eva Dolenc[1☯]*, Damjan Slabe[1☯], Uroš Kovačič[2☯‡]

1 Faculty of Health Sciences, Sanitary Engineering Department, Public Health Division, University of Ljubljana, Ljubljana, Slovenia, 2 Faculty of Medicine, Institute of Pathophysiology, University of Ljubljana, Ljubljana, Slovenia

☯ These authors contributed equally to this work.
‡ Senior author.
* eva.dolenc@zf.uni-lj.si

## Abstract

We aimed to determine the needs and opportunities of older lay people to obtain first aid skills. We determined the level of theoretical knowledge of performing first aid with a structured questionnaire, performed on the sample of 842 adult inhabitants of Slovenia. The method of sampling was balanced by using a system of sampling weights in order to correct deviations in the structure of the sample to the level of the population structure. We also checked their attitude regarding the renewal of first aid knowledge. The level of self-assessment of first aid knowledge and actual theoretical knowledge of proper first aid measures typically decreased with age. The percentage of those who had attended first aid courses at any time was statistically significantly lower among respondents over the age of 60; 38% of elderly respondents thought they needed to renew their first aid knowledge, and 44% would attend a suitable first aid course. None of the 29 European Red Cross and Red Crescent Societies member states that responded have a developed a formally adopted first aid program to train the elderly. A tailored first aid training program for the elderly could be one of the many steps that should be taken to ensure adequate health care for the elderly population.

## Introduction

Although the share of the elderly in the population is constantly increasing, the ageing of the population is not recognized as an illness; rather, it is identified as a health phenomenon [1]. After children, the elderly represent the second most vulnerable group of the population [2, 3]. The rate of hospitalizations after 65 years of age is significantly higher than in other age groups [4, 5]. The number of people with functional decline or disability is increasing, and injuries caused by falls are more frequent among the elderly [6]. Falls represents a major public health problem among the elderly because they lead to premature mortality, loss of independence, and placement in assisted-living facilities [7]. Furthermore, the risk of fatal acute complications of chronic conditions increases with age due to multimorbidity present in more than 50% of elderly people [8]. Frailty, which is significantly associated with morbidity and mortality, is another specific problem of elderly people [9, 10]. Therefore, among the most important

**Data Availability Statement:** All relevant data are within the manuscript and its Supporting Information files.

**Funding:** The author(s) received no specific funding for this work.

**Competing interests:** The authors have declared that no competing interests exist.

target groups for preventive programs for illness alleviation are senior citizens. The key values of health promotion are empowerment of the people to take care of their health, social justice and respect for diversity [11]. Nonetheless, elderly people must take responsibility for their health, a prerequisite being that they are appropriately educated and empowered [4]. Offering support for self-management is one tool for providing a person-centred public-health approach [12]. It consists of providing the elderly with the information, skills and tools that they need to manage their health conditions [1]. Chronic disease self-management programs have been shown to improve a wide range of outcomes among elderly [13]. When first aid needs arise, the elderly are often dependent on self-help or mutual help between relatives. Therefore, promoting first aid education among elderly so they could be self-helpful, appears imperative [14]. Increasing risks for injuries and sudden illnesses are why relevant organizations worldwide emphasize the meaning of knowledge of first aid, including for the elderly [1, 4, 14]. Among the organizations working to raise first aid knowledge are American Heart Association in collaboration with the International Liaison Committee on Resuscitation European Resuscitation Council, St John Ambulance, St Andrew's Ambulance Association, European Committee for First Aid Education, First Aid International etc. Nevertheless, for more than 150 years, first aid has been one of the principal services provided by the Red Cross and the Red Crescent. Each year, Red Cross and Red Crescent National Societies worldwide train more than 15 million people in first aid, and there are currently more than 180.000 active first aid trainers serving their communities, making first aid available for all. In 2015, the International Federation of Red Cross and Red Crescent Societies promoted the role of elderly people with the slogan for World First Aid Day, "First aid and ageing of population", with which a positive role of ageing is emphasized along with the identification of elderly people as an important source of first aid knowledge for society [14]. This study aimed to determine the needs and opportunities of older lay people in Europe to obtain first aid skills. First, we determined the level of theoretical knowledge to perform first aid among laypeople in Slovenia and their attitude regarding the renewal of first aid knowledge. The focus was on the comparison between different age groups. We also explored the first aid courses with adjusted curricula that are offered to elderly people in Europe.

## Methods

### Subjects and procedure

Data collection was performed by The public opinion and Mass Communication Research Centre of the Faculty of Social Sciences of the University of Ljubljana. The sampling process took place in two stages. In the first phase, quotas of telephone numbers were randomly selected from the database of all telephone subscribers (Telekom Slovenia). A representative sample base of telephone subscribers (n = 3355) was formed according to the regional proportions of Slovenia (size of settlement, region). In the second phase of the selection, random adults were selected within the household according to the "last birthday" method. The whole process is based on randomness and ensures that people from the (accessible) population have the same probability of selection. Respondents were acquainted with the purpose of the questionnaire and agreed to participate. Informed consent was obtained. The size of the sample base (n = 3355) was exhausted up to 25.4% (rate of response), which means that 853 surveys were completed by telephone interviews. Telephone interviews were conducted by pre-trained professional interviewers. The authors verified the first aid courses with adjusted curriculum for elders via email and telephone interviews among all (n = 56) International Federation of Red Cross and Red Crescent Societies [15] in Europe. The study was approved by the National Medical Ethic Committee, Ministry of Health and completed in 2018.

## Instruments

Data were collected with a structured questionnaire (see S1 Appendix) which enabled the measurement of some quantitative variables. The questionnaire was designed on the authors' pre-existing empirical research, dealing with various aspects of first aid. The questionnaire was pre-tested and evaluated by expert evaluators; it contained demographic data, and four sets of substantive questions: knowledge of first aid, willingness to give first aid, previous training sessions in first aid, and willingness to attend refresher courses. All questions, except the question about the emergency medical telephone number, were of the closed type.

## Statistical analysis

The demographic data were shown using descriptive statistics. The presentations and statistical analysis of the data are based on a system of sampling weights that reduce the bias in the results due to the method of sampling and conducting the survey [16–18]. Weights correct deviations in the structure of the sample to the level of the population structure, according to the shares that are accessible from statistical sources (Slovenian Public Opinion Survey). The weights were formed by the raking method (by mimicking marginal frequencies) on the variables gender (2 categories), age (4 categories), education (4 categories) and settlement size (4 categories). Not all respondents answered all the questions; therefore, results are shown by numerals and percentages. Dependent variables obtained with substantive questions were expressed as percentages (%) of affirmative or correct answers. The statistically significant differences between the different categories of age groups were determined with the $\chi^2$ square test, followed by appropriate post hoc testing for multiple comparisons. The statistical significance in percentages of correct answers about different questions on injuries in each age group was determined with McNemar's test. Due to multiple comparisons, the level of statistical significance was adjusted ($p < 0.005$). The average ratings of actual knowledge of first aid skills were self-assessed using the Likert scale (from 0—lowest grade to 10—highest grade) and expressed as means ± CI (confidence interval). Statistically significant differences in average ratings among different age groups were determined with one-way ANOVA followed by a post hoc Tukey test with appropriate adjustments of the p-value for multiple comparisons. The significant correlation between age categories and four different time windows since the last renewal of first aid knowledge was assessed using the Spearman correlation coefficient.

## Results

All respondents (n = 842; Table 1) were over 18 years of age, and they came from different age groups. The demographic analysis showed that more than half (57%) all respondents have a secondary school education and come from urban centres (45%). Among those under the age of 31, most (34%, 96/186) had completed high school. In the 31–45 and 46–60 age groups, 38% (68/244) and 30% (53/218) of the respondents, respectively, had a university education. Among the respondents aged over 60, most (43%, 81/196) had completed primary school. In the sample, 9% were (74/842) healthcare professionals. The highest percentage of healthcare professionals (15%, 29/186) was in the age group below 31 years and the lowest in the age groups between 46 and 60 years (11/217) and over 60 years (11/195).

## Assessment of first aid knowledge

The self-assessment results of the knowledge of respondents in individual age categories are shown in Fig 1. The average self-assessment of all respondents was 5.4 ± 1.95 (mean ± SD). The youngest age group rated themselves on an 11-point scale from 0 (worst grade) to 10 (best

**Table 1. Demographic data of all respondents.**

| Sex | MALE | FEMALE | | |
|---|---|---|---|---|
| | 49% | 51% | | |
| **Age (years)** | < 31 | 31–45 | 46–60 | > 60 |
| | 22% | 29% | 26% | 23% |
| **Education** | PRIMARY SCHOOL | SECONDARY SCHOOL | HIGH SCHOOL -GYMNASIUM | HIGHER-EDUCATION* |
| | 22% | 23% | 34% | 21% |
| **Place of residence** | THE COUNTRYSIDE | URBAN CENTRES | LARGEST CITIES IN SLOVENIA# | |
| | 37% | 45% | 18% | |
| **Health care provider** | YES | NO | | |
| | 9% | 91% | | |

Legend:

*B3–1st and 2nd or 3rd cycle of Bologna.

#Ljubljana (approximately 280000 inhabitants), Maribor (approximately 94000 inhabitants).

grade) with the highest mean score 6.3 ± 1.8 of first aid knowledge, which was statistically significantly different (p < 0.05) from all other groups. The group of respondents aged 31 to 45 years also assessed itself with a statistically significantly (p < 0.05) higher mean score than the oldest age group.

Three quarters of all respondents knew the correct emergency telephone number, which was a three-fold larger proportion in comparison with the knowledge of the correct ratio of chest compressions and rescue breaths during resuscitation and about a 1.5 larger proportion compared to the knowledge of the correct measures in case of partial airway obstruction (see Table 2). In the oldest (> 60 years) age group, the proportion of the respondents who knew the emergency telephone number was almost 20% lower than in the youngest age group, which was statistically significantly different ($\chi 2$ = 14.318; p = 0.003). In the oldest group, the proportion of the respondents who knew the correct ratio of chest compressions and rescue breaths in the resuscitation of an adult was four times lower than in the youngest age group

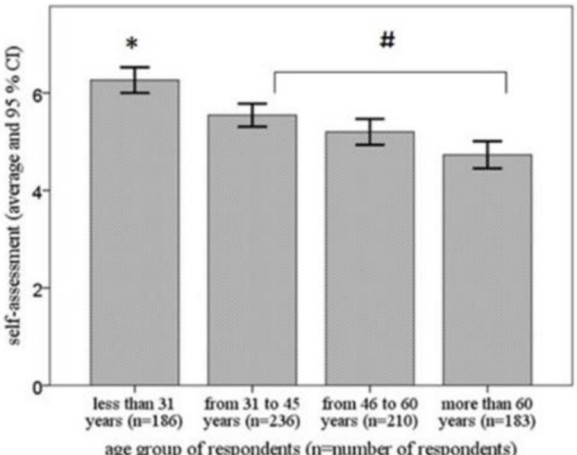

**Fig 1. Self-assessment of the knowledge of first aid assessed by the 11-point Likert scale (from 0 -lowest grade to 10—highest grade).** Legend: Average grades for different age categories of the respondents are expressed as means ± CI (confidence interval). n–number of respondents per in age group; *—statistically significantly (p < 0.05) different when compared with all other groups; #—statistically significantly (p < 0.05) different from the oldest age group.

**Table 2. Proportions of the correct answers to the selected questions about the first aid measures.**

| Questions | Age groups of respondents (years) | | | | All interviewed |
|---|---|---|---|---|---|
| | < 31 | 31–45 | 46–60 | > 60 | |
| Call center telephone number (112)[1] | 84% * | 76% | 73% | 67% | 75% |
| | (156/186) | (184/243) | (159/217) | (132/196) | (631/842) |
| Compression-to-breath ratio (30:2)[2] | 48% # | 25% * | 18% | 12% | 25% |
| | (88/185) | (62/244) | (40/218) | (23/195) | (213/842) |
| Partial airway obstruction[2] | 64% # | 42% | 46% | 42% | 48% |
| | (119/185) | (101/243) | (99/217) | (81/195) | (400/840) |

The percentages and numbers of correct answers according to the number of all respondents in each age group are shown.

*—statistically significantly (p < 0.05) different when compared to the oldest age group;

#—statistically significantly (p < 0.05) different when compared to all other groups; 1-open question, 2-closed question.

and two times lower than in those aged between 31 and 45 years, which were statistically significantly differences ($\chi2 = 72.937$; $p < 0.001$). The proportion of correct answers on the question about the appropriate measures in case of a partial airway obstruction was also statistically significantly lower in the oldest group than in the youngest group ($\chi2 = 27.507$; $p < 0.001$; see Table 2).

When asking about the knowledge of the correct measures for the selected injury cases (Table 3) a statistically significant (p < 0.05) worst result was found with regards to the appropriate care in amputation, where only a quarter of all respondents knew the rule that an amputated limb should never be put directly on ice. In contrast, the proportion of the respondents who answered correctly that people should not take the person with a hip fracture to a healthcare facility by themselves was the largest (82% of all the respondents), which is statistically significantly different (p < 0.05) compared to the knowledge of measures in the cases of other injuries (Table 3). When comparing age-related differences in the proportions of correct answers regarding the measures in different injuries, we found that the older respondents had shown the poorest first aid knowledge in all the selected injury cases (Table 3). Regarding the

**Table 3. Proportions of correct answers to the statements about first aid measures in the selected cases of injuries in the questionnaire.**

| Statement about first aid measures in selected injuries | Age groups of respondents (years) | | | | All interviewed |
|---|---|---|---|---|---|
| | < 31 | 31–45 | 46–60 | > 60 | |
| "Severe bleeding is stopped by direct pressure on the bleeding site" | 66% | 54% | 48% * | 50% * | 54% |
| | (122/186) | (132/243) | (103/217) | (99/196) | (456/842) |
| "The amputated part is not directly placed on ice" | 32% | 24% | 24% | 19% * | 25% + |
| | (60/186) | (57/243) | (52/217) | (38/196) | (207/842) |
| "In case of bone and joint injury, the person should not eat and drink" | 51% | 69% # | 54% | 45% | 58% |
| | (94/186) | (168/243) | (118/217) | (105/195) | (485/841) |
| "After the fracture of the hip, the injured person can be taken to a health facility" A | 81% | 89% $ | 86% $ | 70% | 82% + |
| | (149/185) | (216/243) | (187/218) | (135/194) | (687/840) |

The percentages and numbers of correct answers according to the number of all respondents in each age group are shown.

A—the measure is incorrect (the correct answer is NO);

*—statistically significantly (p < 0.05) different when compared to the youngest (< 31 years old) age group;

#—statistically significantly (p < 0.05) different when compared to all other groups;

$—statistically significantly (p < 0.05) different when compared to the oldest (> 60 years old) age group; +—statistically significantly (p < 0.05) different when compared to other statements about selected injuries.

question about the correct measure in amputation, a statistically significantly ($\chi^2 = 8.974$; $p = 0.03$) smaller percentage of correct answers was found in the oldest group of respondents compared to the youngest age group (Table 3). The rule on transporting a person with a hip fracture is also least known in the oldest age group (70%), which was statistically significantly ($\chi^2 = 30.135$; $p < 0.001$) lower than in the two age groups with middle-aged respondents (aged between 31 and 60 years), but not statistically significantly different from the youngest group. There were also no statistically significant differences ($\chi^2 = 0.785$; $p = 0.376$; Table 3) between the oldest and the youngest age groups regarding knowledge of the rule of eating and drinking in case of a major injury (i.e., fracture or sprain). Regarding the knowledge about stopping the bleeding we found a statistically significantly ($\chi^2 = 14.761$; $p = 0.002$) lower proportion of correct answers in both groups of respondents over the age of 45 when compared to the youngest group (see Table 3).

## Willingness to give first aid to injured person and attitude towards the first aid training

Three quarters of all respondents answered that they would help if a person would need first aid (Table 4). Concerning this question, we found no statistically significant differences ($\chi2 = 2.185$; $p = 0.535$) between the age groups. The percentage of respondents who had already undergone first aid training was statistically significantly ($\chi2 = 48.174$; $p < 0.001$) lower in the oldest age group than the other three age groups. Of the 150 respondents over the age of 60 who had already undergone first aid training, 57 felt that they should renew their knowledge, which was statistically significantly ($\chi2 = 41.375$; $p < 0.001$) less than in the other age groups (see Table 4). The proportion of those who would attend a first aid course, if they would be timely informed about the start of such a course (Table 4), was almost one-half higher in all three groups under 60 years of age than in the group of respondents older than 60 years, which was statistically significantly different ($\chi2 = 27.512$; $p < 0.001$). Elderly people who thought that they should renew first aid knowledge indicated different reasons for not doing so. Almost one third (31%) stated that there is no training offer, 9% of them expressed financial constraints and 14% of the elderly exposed their lack of interest in such a course, which was a statistically significantly ($p < 0.05$) larger share than among younger respondents. In addition, 16% of them do not feel capable, which was statistically significantly larger share ($p < 0.05$) than among younger respondents, where this proportion was 3%. Among the oldest

**Table 4. Willingness to give first aid to the injured person and interest in renewing knowledge of first aid skills.**

| Questions | Age groups of respondents (years) | | | | All interviewed |
|---|---|---|---|---|---|
| | < 31 | 31–45 | 46–60 | > 60 | |
| Would you be willing to offer first aid? | 78% | 72% | 73% | 75% | 74% |
| | (145/186) | (175/243) | (158/217) | (145/195) | (623/841) |
| Have you been educated in first aid in the past? | 94% | 96% | 91% | *77% | 90% |
| | (174/186) | (234/244) | (199/218) | (151/196) | (758/844) |
| Should you renew the knowledge of first aid?[A] | 62% | 70% | 63% | *38% | 60% |
| | (108/174) | (162/231) | (126/199) | (57/150) | (453/754) |
| I would attend a first aid course. | 64% | 66% | 64% | *44% | 60% |
| | (119/186) | (161/244) | (139/217) | (85/194) | (504/841) |

The shares (%) and the number of affirmative answers are shown in relation to the total number of respondents by individual age categories.

[A]–only those who had already attended the first aid course;

*—statistically significantly ($p < 0,001$) different from other age groups.

age group 17% don't have enough time, which was a statistically significantly smaller share (p < 0.05) than among younger respondents where the shares were 35–40%.

The highest proportion of respondents in the present study had undergone their last first aid training more than 10 years ago (57% or 431 out of 756 respondents). In contrast, slightly less than 14% (104 out of 756 respondents) had last attended a first aid training less than a year ago (Table 5). The correlation between the categories of age and the number of years that have elapsed since the last first aid training is statistically significantly positive (p < 0.001) and moderately strong in terms of the Spearman correlation coefficient (ρ, ro = 0.548). In the group of respondents aged 46–60 (n = 199), the proportion of those who had undergone their last first aid training more than 10 years previously was 76%, while in the oldest group (over 60 years of age; n = 150), this proportion was the highest (91%) among all age groups (see also Table 5). Of those younger than 31 years of age (n = 173), 72% had undergone their last first aid training less than five years previously, and only 7% had undergone their last training more than 10 years previously (see also Table 5). Among the respondents who had undergone the course in the last year, more than half (55 out of 104 respondents) were younger than 31 years, and only three respondents were older than 60 years (Table 5).

## The first aid courses offered to the elderly

Of the 54 Member States of the International Federation of Red Cross and Red Crescent Societies [15] in Europe, 13 Member States immediately responded to the letter sent via e-mail: Austria, Belgium, Great Britain, Croatia, Cyprus, Finland, France, Italy, Luxembourg, Monaco, Serbia, Slovenia, Turkey; the remaining 16 responded to the additional telephone survey: Bosnia and Herzegovina, Czech Republic, Denmark, Estonia, Greece, Iceland, Ireland, Latvia, Liechtenstein, Montenegro, the Netherlands, Norway, Portugal, Romania, Slovakia, Sweden. None of the 29 Member States that responded has a first aid program tailored to train the elderly.

## Discussion

### Assessment of first aid knowledge among the elderly

In the present research, we found that the level of self-assessment of first aid knowledge among the representative sample of 842 adult inhabitants of Slovenia statistically significantly

**Table 5. Time period from the last first aid training course.**

| Time period (years) from last first aid training course | Age groups of respondents (years) | | | |
|---|---|---|---|---|
| | < 31 | 31–45 | 46–60 | > 60 |
| | (n = 173) | (n = 234) | (n = 199) | (n = 150) |
| < 1 (N = 104) | 53% | 31% | 14% | 3% |
| | 55/104 | 32/104 | 14/104 | 3/104 |
| 2–5 (N = 149) | 47% | 35% | 14% | 4% |
| | 70/149 | 52/149 | 21/149 | 6/149 |
| 5–10 (N = 72) | 50% | 26% | 17% | 7% |
| | 36/72 | 19/72 | 12/72 | 5/72 |
| > 10 (N = 431) | 3% | 30% | 35% | 32% |
| | 12/431 | 131/431 | 152/431 | 136/431 |

The shares (%) of respondents are shown by age groups in relation to the total number of respondents in individual time windows that have elapsed since the last first aid training. N—number of all in each time period; n–number of all per age group.

decreases with age. The average rate of self-assessment was 25% lower in the group of those older than 60 years compared to younger respondents. In addition, for all seven questions about first aid measures in selected injury cases, the proportion of correct answers was the lowest in the group of the oldest respondents. Furthermore, in six out of the seven selected first aid measures, the proportion of correct answers in the oldest group was statistically significantly lower than the group of the youngest respondents and in three out of the seven questions than in the group aged between 31 and 45 years. In contrast, the result in the oldest group was statistically significantly worse than in the group aged 45 to 60 years in only one of the seven questions. Among other factors, knowledge about the basic cardiopulmonary (CPR) procedures in our survey is also statistically significantly worse among the older respondents than the younger ones. It was shown that patients with out-of-hospital cardiac arrest in 2020 were elderly with physical limitations and a high burden of hypertension and diabetes [19]. Most cardiac arrest victims are in their late sixties and half of them are witnessed by a family member or friend who is usually over the age of 55 [20]. Therefore, elderly people are the population that needs first aid knowledge the most. Concerning the knowledge of resuscitation procedures, 51% of the older respondents in the study of Brinkrolf [21] believed that they knew how CPR was performed, which was worse than the group of younger people. Poor knowledge of CPR could explain the results of Dobbie et al. [22], who found that the older the person is, the less likely he/she is confident to administer CPR with or without instructions from an emergency call handler. In contrast, it is encouraging that the great majority of all respondents in our survey answered that we should not drive a person with a hip fracture to a healthcare facility by ourselves. Even so, the oldest respondents were the least successful in answering this question. Of note, bone injuries (especially hip fractures) due to falls are common injuries among the elderly [23, 24]; 30% and 50% of persons over 60 and 80 years old, respectively, fall at least once per year, and falls are the main risk factor for fractures (1). Approximately 5–10% of older persons sustain hip fractures and wrist injuries requiring hospital treatment [25, 26] and 40% of women older than 50 years can expect to have a hip, vertebral or forearm fractures in their remaining lifetime [27]. The knowledge of the communication protocol (telephone number, use of speakerphone feature on the mobile phone, etc.) for emergency medical assistance among laypeople is also important because of the increasing role of dispatchers of performing basic resuscitation procedures via mobile phone instructions. We found that 67% of the respondents older than 60 years know the correct emergency telephone number, which is in line with a similar survey conducted in Germany, which found that 82% of interviewees under 65 years of age knew the correct emergency number [21]. However, Birkenes and colleagues [28] found that only a third of those over 50 years of age know how to activate the speakerphone feature on their mobile phone, while others spend too much time activating it or are unable to activate it. Notably, Wu et al. [29] found that dispatcher-initiated telephone cardiopulmonary resuscitation was independently associated with increased survival and favorable functional outcome after out-of-hospital cardiac arrest.

## The needs of elderly to acquire knowledge and skills on tailored first aid training

Inadequate education in first aid knowledge could be a reason why older people are less confident of their first aid knowledge and have poorer first aid knowledge than younger individuals. Accordingly, we showed that the oldest respondents attended first aid training at any time in the past less frequently than the younger and that among the older respondents, more time has passed since their last first aid training. It has been pointed out that first aid training participants have more theoretical and practical knowledge of the measures [30, 31]. Furthermore,

they help an injured or suddenly ill person more frequently than the persons who did not attend a first aid training or if much time has passed since they did [32–35]. Notably, Brinkrolf and colleagues [21] also suggested that less frequent implementation of CPR outside the hospital by the eyewitness is associated with age-related decline in CPR knowledge. According to the results of our study, as many as one-quarter of all respondents in our survey decided they would not give first aid to a person in need of medical assistance. Interestingly, there was no statistically significant difference between younger and older people on this issue, indicating that age does not significantly affect ethical principles and a sense of empathy for one's fellow human being. Nevertheless, the most likely reason for abandoning first aid provision is the fear of wrong action (inappropriate first aid), harming the patient and possible legal liability [32, 36, 37]. This indicates that poor knowledge increases the fear of wrong actions, which deters one from taking any action at all. To summarize, our research results are in accordance with the similar German survey [21] indicating the need to better educate the elderly population on the knowledge of first aid measures.

In our survey, the percentage of those that thought they needed to refresh their first aid knowledge and attend a suitable first aid course was statistically significantly lower in the oldest group, in which we found 38% of those who thought that they needed to renew their first aid knowledge and 44% who would attend a first aid course. One of the key characteristics of the elderly seems to be their great diversity compared to the younger population [1], which should be taken into account when performing any activity in this group, including first aid training. Some obstacles in the field of first aid training also arise due to forgetting the content and the knowledge becoming obsolete. As memory and the ability to learn on average decrease with age [38], the knowledge of first aid should be renewed in older population even more frequently. However, due to the poorer psychophysical condition [39] and reduced ability to concentrate of the elderly [40], first aid refresher courses should be shorter and more focused. Notably, Vaillancourt [20] identified similar key barriers (time commitment, cost, physical limitations) for CPR training and performance in a purposive sample of older individuals as we did. Therefore, it is understandable that it was suggested that the CPR curriculum should be tailored to the target audience and kept as simple as possible [41, 42]. Course organizers have to plan their courses in a flexible way, allowing a shorter duration for target groups with different backgrounds and more hands-on time for lay rescuers. Nevertheless, older people can also learn and implement first aid measures [43–46]. Therefore, it is also necessary to regulate first aid training for the elderly [21, 28, 43–47]. In this regard, it is interesting that among the 54 surveyed International Federation of Red Cross and Red Crescent Societies [15], we did not find any offers of aid courses aimed at the elderly, even though both organizations [1, 14, 48] emphasized the priorities for raising the health literacy (including first aid knowledge) of the elderly. Accordingly, almost one third of older respondents in our survey stated that there is no training offer. As an at-risk population, the elderly are an important target group for the systematic planning of prevention programs. Examples from multiple countries have shown that the integration of different innovations into practice depends on a harmonized approach of the science, technology and policy environment [13, 49]. Of course, appropriate health system responses must be country-specific and respect local context, but cross-country comparisons and best practice cases can often help initiate and refine policy responses to mitigate the challenges of ageing [50].

## Limitations and recommendations for further research

One of the limitations of our research is that we only investigated theoretical knowledge of first aid. Behavior in a real situation could be different. In a real situation of cardiac arrest,

Park et al. [51] found that people over 60 perform CPR at a lower quality. Notably, Takei and colleagues [52] confirmed that older eyewitnesses are less likely to perform CPR in reality. The reasons for and sources of acquiring FA knowledge considering other opportunities and possibilities (i.e. in the mass media, at the family doctor) were also not explored in more detail in the present study. An additional limitation of our research is also that the questionnaire was focused to be made on a representative sample of the whole population of Slovenia, but not on a representative sample of the elderly, although the total number of respondents older than 60 was not small. This limits the further research of differences in knowledge and attitudes towards FA among the elderly, which may be due to the great diversity in this population. We are aware that in addition to the Red Cross, there are many other organizations that offer first aid education. With all due respect to their work, we did not check the offers of first aid courses for the elderly among other organizations, which is a limitation of our research.

## Conclusion

The health policy must include a healthy ageing program that systematically promotes healthy living throughout a person's life span [4]. However, if a disease or injury develops, preventative strategies are not sufficient, and the health policy must also include continuous education of elderly lay people to be better informed about the disease. The latter also includes first aid in cases of injuries or sudden diseases. The results of our survey show that the level of self-assessment of first aid knowledge and the actual theoretical knowledge of proper first aid measures decreased with age. In addition, none of the 29 EIRCS have developed a formally adopted first aid program to train the elderly. An innovative and tailored first aid training program for the elderly, which would be at least partially personalized [53], could be one of the steps that should be taken to adequately ensure public health for the older population. Such a program, which would be in line with the priorities of the World Health Organization, International Federation of Red Cross and Red Crescent Societies and other competent organization in this field, would adequately empower an elderly person to achieve the goals of first aid in the case of injury or sudden illness.

## Supporting information

**S1 Appendix. First aid questionnaire.**
(DOCX)

**S1 Data.**
(XLSX)

## Author Contributions

**Conceptualization:** Damjan Slabe.

**Formal analysis:** Uroš Kovačič.

**Writing – original draft:** Eva Dolenc.

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
