## [Decision Letter · Decision Letter 0]

19 Mar 2021

PONE-D-21-04651

The needs and opportunities of older laypeople in Europe to acquire first aid skills

PLOS ONE

Dear Dr. Dolenc,

Thank you for submitting your manuscript to PLOS ONE. After careful consideration, we feel that it has merit but does not fully meet PLOS ONE’s publication criteria as it currently stands. Therefore, we invite you to submit a revised version of the manuscript that addresses the points raised during the review process.

We look forward to receiving your revised manuscript.

Kind regards,

Filipe Prazeres, MD, MSc, Ph.D.

Academic Editor

PLOS ONE

Journal Requirements:

2. Please include additional information regarding the survey or questionnaire used in the study and ensure that you have provided sufficient details that others could replicate the analyses.

For instance, if you developed a questionnaire as part of this study and it is not under a copyright more restrictive than CC-BY, please include a copy, in both the original language and English, as Supporting Information.

3. Please provide additional details regarding participant consent. In the ethics statement in the Methods and online submission information, please ensure that you have specified (i) whether consent was informed and (ii) what type you obtained (for instance, written or verbal, and if verbal, how it was documented and witnessed).

If your study included minors, state whether you obtained consent from parents or guardians. If the need for consent was waived by the ethics committee, please include this information.

Reviewers' comments:

Reviewer's Responses to Questions

**Comments to the Author**

1. Is the manuscript technically sound, and do the data support the conclusions?

Reviewer #1: Yes

Reviewer #2: No

2. Has the statistical analysis been performed appropriately and rigorously? 

Reviewer #1: Yes

Reviewer #2: No

3. Have the authors made all data underlying the findings in their manuscript fully available?

Reviewer #1: Yes

Reviewer #2: Yes

4. Is the manuscript presented in an intelligible fashion and written in standard English?

Reviewer #1: Yes

Reviewer #2: Yes

5. Review Comments to the Author

Reviewer #1: Dear Authors,

I had the opportunity to review your very interesting manuscript. It is well written and thought-out, and adds an important piece of information to already existing literature. However, a few points need to be addressed before reconsidering your work for a broader audience:

Introduction

-) Lines 45-47: This reads very vague and too generalized, please rephrase or extend.

-) Lines 48-57: Why do you emphasize the Red Cross so much when first aid courses are provided by many different kinds of organisations and instutions (surely also in your country)? At least other sources of first aid education must be mentioned.

Methods

-) Lines 66-67: I do not see why this information is relevant?

-) Lines 69-74: This paragraph reads a little confusing, please clarify for a potential reader.

Results

-) Overall: Maybe you could shorten the Results section a bit? It reads very long and a potential reader might lose interest.

-) Lines 241-249: What about other organisations / institutions potentially offering first aid courses for elderly people?

Discussion

-) Your disussion is quite long and not structured visually. Please consider using subheadings in order to structure the section more and make it more apealing to a potential reader.

-) Line 266 "Therefore...": This statement seems too general, please rephrase.

Reviewer #2: Dear authors,

thank you for this manuscript.

There are a few pionts that need to be clarified.

1. You data solely consists of information from slowenia, how can you talk about the level of knowledge in Eurpoe in general?

2. You are talking about needs in Europe for first aid courses with adjusted curricula for people over 60 years of age. Why is this nessessary?

3. Based on you questionaire, slightly less people ofer 60 are interested in a first aid couse than younger people in slowenia. You did not ask, if those questioned people over 60 years of age would be interested in a class with adjusted curricula.

4. Additionally you solely questioned 296 people over 60 years of age. How is it possible to gereralize you thesis with such a small group. Please explain and show statistics.

5. Please show how the questioned number of 842 people to interview was calculated statistically. Why did you not question more people. How where the questioned people picked?

6. What was the study period? Who performed the telefone interviews, how did you recieve the telefone numbers?

7. You questions on which your conclusion is based where not adjusted for the elderly, please explain.

Thank you very much.

6. PLOS authors have the option to publish the peer review history of their article (what does this mean?). If published, this will include your full peer review and any attached files.

Reviewer #1: No

Reviewer #2: No

---

## [Author Response · Author response to Decision Letter 0]

30 Jun 2021

Journal Requirements:

 A: Thank you. We have checked your style requirements and made some corrections.

2. Please include additional information regarding the survey or questionnaire used in the study and ensure that you have provided sufficient details that others could replicate the analyses.

For instance, if you developed a questionnaire as part of this study and it is not under a copyright more restrictive than CC-BY, please include a copy, in both the original language and English, as Supporting Information.

A: Based on your suggestion, we have included questionnaire as Appendix A.

3. Please provide additional details regarding participant consent. In the ethics statement in the Methods and online submission information, please ensure that you have specified (i) whether consent was informed and (ii) what type you obtained (for instance, written or verbal, and if verbal, how it was documented and witnessed).

A: We have performed a telephone survey from a database based on the publicly available telephone numbers. Respondents were acquainted with the purpose of the questionnaire and verbally agreed to participate. It was a call to a household – the "witness" was usually another member of the household.

If your study included minors, state whether you obtained consent from parents or guardians. If the need for consent was waived by the ethics committee, please include this information.

A: Our research is not about medical or other personal data. We researched opinions on a specific topic and questions about the knowledge. Data were anonymous. We briefed the respondents about the purpose of the survey and data collection. Data collection was performed by the Public Opinion and Mass Communication Research Centre of the Faculty of Social Sciences of the University of Ljubljana. Pre-trained professional interviewers conducted the telephone interviews. The National Medical Ethics Committee, Ministry of Health, approved the study. 

Reviewers' comments:

Reviewer's Responses to Questions

Comments to the Author

1. Is the manuscript technically sound, and do the data support the conclusions?

Reviewer #1: Yes

Reviewer #2: No

2. Has the statistical analysis been performed appropriately and rigorously?

Reviewer #1: Yes

Reviewer #2: No

3. Have the authors made all data underlying the findings in their manuscript fully available?

Reviewer #1: Yes

Reviewer #2: Yes

4. Is the manuscript presented in an intelligible fashion and written in standard English?

Reviewer #1: Yes

Reviewer #2: Yes

5. Review Comments to the Author

 

Reviewer #1: Dear Authors,

I had the opportunity to review your very interesting manuscript. It is well written and thought-out, and adds an important piece of information to already existing literature. However, a few points need to be addressed before reconsidering your work for a broader audience:

A: Thanks for your comments. We are glad that the topic about first aid for elderly you have also recognized interesting. We have followed all your suggestions. We believe that with your help our research has gained in value.

Introduction

-) Lines 45-47: This reads very vague and too generalized, please rephrase or extend.

A: Thank you for your suggestion. We have extended the paragraph with an in-depth explanation and transferred it to the field of first aid (lines 48-53).

-) Lines 48-57: Why do you emphasize the Red Cross so much when first aid courses are provided by many different kinds of organisations and instutions (surely also in your country)? At least other sources of first aid educationmust be mentioned must be mentioned.

A: We agree with your opinion that first aid (FA) courses are provided by many different kinds of organisations and institutions. We have therefore added some of them and made a brief presentation. Nevertheless, it should be noted that IFRC is one of the few organizations, if not the only one, that operates in all European countries (lines 55-59 and 399-401 as limitations).

Methods

-) Lines 66-67: I do not see why this information is relevant?

A: We agree that this information is not required at this point. Therefore, we have deleted it. In addition, in this paragraph, we have added a more detailed explanation of the methodological procedure, which was required also by the second reviewer (lines 78-87).

-) Lines 69-74: This paragraph reads a little confusing, please clarify for a potential reader.

A: Thank you for your comment. In this paragraph, we have added a more detailed explanation of the methodological procedure (lines 78-87 and 101-109).

Results

-) Overall: Maybe you could shorten the Results section a bit? It reads very long and a potential reader might lose interest.

A: Based on your suggestion, we have shortened the Results section, and tried to make it more interesting for the potential reader.

-) Lines 241-249: What about other organisations / institutions potentially offering first aid courses for elderly people?

A: We agree with your opinion that many different kinds of organisations and institutions provide first aid courses. Therefore, we have added some of them and briefly represented them in the Introduction section (lines 55-59). However, we checked the formally offered adopted courses for the elderly only at the IFRCS because it is one of the few, if not the only, that operates in all European countries. In addition, the authors of the manuscript have been professionally involved in the first aid education for more than 20 years, and we do not know of any organization in Slovenia (and also in Europe) that would have developed educational programs of FA focused on the elderly (personal data). Nevertheless, we agree at some point with the reviewer, that's why we added a new section (Limitations and recommendations for further research) that includes our research limitation.

Discussion

-) Your disussion is quite long and not structured visually. Please consider using subheadings in order to structure the section more and make it more apealing to a potential reader.

A: Based on your suggestion, we subtitled the Discussion section in order to structure the section more.

-) Line 266 "Therefore...": This statement seems too general, please rephrase.

A: We changed the too general interpretation and supported it with an argument (lines 299-304). Thank you.

Reviewer #2: Dear authors, thank you for this manuscript. There are a few pionts that need to be clarified.

1. You data solely consists of information from slowenia, how can you talk about the level of knowledge in Eurpoe in general?

A: Indeed, our questionnaire about FA knowledge was conducted among the Slovenian population. However, the research on the offered first aid training among the elderly was done among the European members of the International Federation of Red Cross and Red Crescent Societies. Besides, in the discussion section we compared the knowledge found with some related research in Europe. That is why the title of the manuscript was more general, considering “The needs and opportunities… in Europe”. Nevertheless, we agree with your comment that the title of the manuscript might be misleading. To avoid ambiguity, we omitted the term "in Europe" from the title and from the abstract.

2. You are talking about needs in Europe for first aid courses with adjusted curricula for people over 60 years of age. Why is this necessary?

A: Based on your comment, we found that we did not explain this problem well enough. Therefore, in the subsection The needs of elderly to acquire knowledge and skills on tailored first aid training (see Discussion), we have added some reasons why adjusted curricula for people over 60 is needed. Thank you for your suggestion.

3. Based on you questionaire, slightly less people over 60 are interested in a first aid couse than younger people in slowenia. You did not ask, if those questioned people over 60 years of age would be interested in a class with adjusted curricula.

A: The questions were prepared in advance, so, unfortunately, we did not really ask this question directly. However, we did ask the people who thought that they should renew first aid knowledge about reasons for not doing so. We added those reasons to the Results section to the paragraph above Table 4. In addition, we made a comment in the discussion and compared it with related findings, arguing the need for a course with adjusted curricula.

4. Additionally you solely questioned 296 people over 60 years of age. How is it possible to gereralize you thesis with such a small group. Please explain and show statistics.

Please show how the questioned number of 842 people to interview was calculated statistically. Why did you not question more people. How where the questioned people picked?

A: In the revised manuscript, we described this methodological issue more in detail. In the Methods we have added a more detailed explanation of the methodological procedure.

The sampling process took place in two stages. In the first phase, quotas of telephone numbers were randomly selected from the database of all telephone subscribers (Telekom Slovenia). A representative sample base of telephone subscribers (n = 3355) was formed according to the regional proportions in Slovenia (size of settlement, region). In the second phase of the selection, random adults were selected within the household according to the “last birthday” method. The whole process is based on randomness and ensures that people from the (accessible) population have the same probability of selection. The size of the sample base (n = 3355) was exhausted up to 25.4% (rate of response), which means that 853 surveys were completed by telephone interviews. The presentations and statistical analysis of the data are based on the system of sampling weights that reduce the bias in the results due to the method of sampling and conducting the survey. Weights correct deviations in the structure of the sample to the level of the population structure, according to the shares that are accessible from statistical sources (Slovenian Public Opinion Survey). The weights were formed by the raking method (by mimicking marginal frequencies) on the variables gender (2 categories), age (4 categories), education (4 categories), and settlement size (4 categories).

6. What was the study period? Who performed the telefone interviews, how did you recieve the telefone numbers?

A: The above-mentioned constructive methodological issues were discussed in more appropriate and in more detail in the Methods (see also above).

7. You questions on which your conclusion is based where not adjusted for the elderly, Please explain.

A: The observation of the reviewer is correct. Indeed, that was also our intention. We would like to emphasize that our conclusions are based on comparison between different age groups. Therefore, we asked questions to the general adult population and then statistically significantly compare different age groups. Groups of different ages were properly balanced using the system of sample weights (see above). Thank you.

Thank you very much.

6. PLOS authors have the option to publish the peer review history of their article (what does this mean?). If published, this will include your full peer review and any attached files.

Do you want your identity to be public for this peer review? For information about this choice, including consent withdrawal, please see our Privacy Policy.

Reviewer #1: No

Reviewer #2: No

---

## [Decision Letter · Decision Letter 1]

28 Jul 2021

The needs and opportunities of older laypeople to acquire first aid skills

PONE-D-21-04651R1

Dear Dr. Dolenc,

We’re pleased to inform you that your manuscript has been judged scientifically suitable for publication and will be formally accepted for publication once it meets all outstanding technical requirements.

Kind regards,

Filipe Prazeres, MD, MSc, Ph.D.

Academic Editor

PLOS ONE

Additional Editor Comments (optional):

Reviewers' comments:

Reviewer's Responses to Questions

**Comments to the Author**

1. If the authors have adequately addressed your comments raised in a previous round of review and you feel that this manuscript is now acceptable for publication, you may indicate that here to bypass the “Comments to the Author” section, enter your conflict of interest statement in the “Confidential to Editor” section, and submit your "Accept" recommendation.

Reviewer #1: All comments have been addressed

Reviewer #2: All comments have been addressed

2. Is the manuscript technically sound, and do the data support the conclusions?

Reviewer #1: Yes

Reviewer #2: Yes

3. Has the statistical analysis been performed appropriately and rigorously? 

Reviewer #1: Yes

Reviewer #2: Yes

4. Have the authors made all data underlying the findings in their manuscript fully available?

Reviewer #1: Yes

Reviewer #2: Yes

5. Is the manuscript presented in an intelligible fashion and written in standard English?

Reviewer #1: Yes

Reviewer #2: Yes

6. Review Comments to the Author

Reviewer #1: Dear authors,

Thank you for addressing all comments, I think that the manuscript has substantially improved. Still, please check for minor spelling and grammar mistakes.

Reviewer #2: Thank you for adressing all of my remarks.

I have no further questions. Nice work and important topic.

7. PLOS authors have the option to publish the peer review history of their article (what does this mean?). If published, this will include your full peer review and any attached files.

Reviewer #1: No

Reviewer #2: No

---

## [Editor Report · Acceptance letter]

1 Oct 2021

PONE-D-21-04651R1 

The needs and opportunities of older laypeople to acquire first aid skills 

Dear Dr. Dolenc:

I'm pleased to inform you that your manuscript has been deemed suitable for publication in PLOS ONE. Congratulations! Your manuscript is now with our production department. 

Kind regards, 

on behalf of

Prof. Filipe Prazeres 

Academic Editor

PLOS ONE